# Performance characteristics and operational feasibility assessment of a CRISPR based tata MD CHECK diagnostic test for SARS-CoV-2 (COVID-19)

**Shubhada Shenai**[1]*, **Gita Nataraj**[2☉], **Minal Jinwal**[2☉], **Akhil S. Thekke Purakkal**[1], **Rajashree Sen**[1], **Nayana Ingole**[2], **Trupti Mathure**[2], **Sanjay Sarin**[1], **Sarabjit S. Chadha**[1]

**1** FIND, New Delhi, India, **2** Department of Microbiology, Seth GSMC, & KEM Hospital, Mumbai, Maharashtra, India

☉ These authors contributed equally to this work.

* Shubhada.shenai@finddx.org

**Data Availability Statement:** All relevant data for this study outcome are within the manuscript and its Supporting Information files.

## Abstract

### Background

Tata MD CHECK SARS-CoV-2 kit 1.0, a CRISPR based reverse transcription PCR (TMC-CRISPR) test was approved by Indian Council of Medical Research (ICMR) for COVID-19 diagnosis in India. To determine the potential for rapid roll-out of this test, we conducted performance characteristic and an operational feasibility assessment (OFA) at a tertiary care setting.

### Intervention

The study was conducted at an ICMR approved COVID-19 RT-PCR laboratory of King Edward Memorial (KEM) hospital, Mumbai, India. The TMC-CRISPR test was evaluated against the gold-standard RT-PCR test using the same RNA sample extracted from fresh and frozen clinical specimens collected from COVID-19 suspects for routine diagnosis. TMC-CRISPR results were determined manually and using the Tata MD CHECK application. An independent agency conducted interviews of relevant laboratory staff and supervisors for OFA.

### Results

Overall, 2,332 (fresh: 2,121, frozen: 211) clinical specimens were analysed of which, 140 (6%) were detected positive for COVID-19 by TMC-CRISPR compared to 261 (11%) by RT-PCR. Overall sensitivity and specificity of CRISPR was 44% (95% CI: 38.1%-50.1%) and 99% (95% CI: 98.2%-99.1%) respectively when compared to RT-PCR. Discordance between TMC-CRISPR and RT-PCR results increased with increasing Ct values and corresponding decreasing viral load (range: <20% to >85%). In the OFA, all participants indicated no additional requirements of training to set up RT PCR. However, extra post-PCR steps such as setting up the CRISPR reaction and handling of detection strips were time

**Funding:** This work was supported, in whole or in part, by the Bill & Melinda Gates Foundation #INV-025422. No additional external funding was received for this study.

**Competing interests:** The authors have declared that no competing interests exist.

consuming and required special training. No significant difference was observed between manual and mobile app-based readings. However, issues such as erroneous results, difficulty in interpretation of faint bands, internet connectivity, data safety and security were highlighted as challenges with the app-based readings.

## Conclusion

The evaluated version-Tata MD CHECK SARS-CoV-2 kit 1.0 of TMC-CRISPR test cannot be considered as an alternative to the RT-PCR. There is a definite scope for improvement in this assay.

## Introduction

Coronavirus disease 2019 (COVID-19) pandemic highlighted the importance of early diagnosis of severe acute respiratory syndrome coronavirus-2 (SARS-CoV-2) [1–3]. To detect SARS-CoV-2 infection, confirmation of viral antigen or viral ribo-nucleic acids (RNA) in the respiratory sample is recommended. Rapid antigen tests are easy to perform at peripheral settings with rapid turn-around time of about 30 minutes. However, they vary in sensitivity and might miss approximately 15–50% cases (false negatives) requiring retesting with the reverse transcription real time polymerase chain reaction (RT-PCR) [4]. In India, the RT-PCR test is limited to centralised laboratories approved by the Indian Council of Medical Research (ICMR) because of the requirement for sophisticated laboratory equipment, expensive reagents, and skilled personnel. To overcome these complexities, the Council of Scientific and Industrial Research and Tata Medical and Diagnostic Ltd. jointly developed a clustered regularly interspaced short palindromic repeats (CRISPR) based diagnostic test Tata MD CHECK SARS-CoV-2 kit 1.0, (TMC-CRISPR) as a simple and cost-effective alternative that received ICMR approval for diagnosis of COVID-19 under emergency use in October 2020 [5].

TMC-CRISPR test is an in-vitro diagnostic test that uses reverse transcription PCR for the amplification of spike protein (S) gene target followed by CRISPR-CAS9 technology to detect SARS-CoV-2 RNA in human respiratory samples [6]. The amplified product is bound by CRISPR-Cas9 based gRNA specifically generated for S target sequences. The complex is then detected by a method known as FnCas9 Editor Linked Uniform Detection Assay which uses lateral flow strips to show the presence or absence of the target sequences as a visible band [6]. The Tata MD CHECK mobile-application (app) measures the intensity of the band formed on the Lateral Flow strip (LFS) and an artificial intelligence (AI) logic classifies the result into positive and negative [6]. An operational feasibility assessment (OFA) and performance evaluation of TMC-CRISPR test in a reference laboratory would inform implementers of the operating conditions, human resource (HR) requirements, user friendliness, quality control needs and other post-launch requirements, and support rapid roll-out of this test to strengthen the diagnosis of SARS-CoV-2. However, such an assessment has not been done in the past. To address this gap, FIND, in collaboration with the virus research and diagnostic laboratory (VRDL) at King Edward Memorial (KEM) Hospital conducted an independent assessment to estimate the diagnostic accuracy and operational feasibility of implementing the TMC-CRISPR test in a RT-PCR section of VRDL.

## Material and methods

This observational study was conducted at the ICMR approved COVID-19 RT-PCR laboratory of KEM hospital, Mumbai, India during July-October 2021. The study received ethics

committee approval from KEM internal institutional review board [Institutional Ethics Committee (IEC)-II relating to biomedical and health research (BHR), Seth GS Medical College and KEM Hospital, Mumbai, Maharashtra, India]. Technical staff performing RT-PCR at KEM VRDL, was trained by manufacturer's technical team to process the clinical specimens and interpret the results using a manual technique as well as through an AI based mobile application developed specifically for the index test.

RNA extraction of clinical specimens including nasopharyngeal specimens (NPS), throat swab (TS), nasal swab (NS) and endotracheal aspirates collected from COVID-19 suspects was carried out manually using Qiagen QIAamp Viral RNA Kit. The same RNA extract was used to perform routine RT-PCR for clinical diagnosis, and to conduct this study. During the study duration, for routine testing of COVID-19 samples and for clinical decision-making, the laboratory used ICMR approved COVIDsure RT-PCR kits, received from the state/or central government.

## Evaluation of diagnostic accuracy

The diagnostic accuracy study followed a prospective, single blinded design. The performance of the index test was primarily validated in upper respiratory tract specimens extracted from COVID-19 suspects who presented at the study site for testing [7, 8]. To assess diagnostic accuracy of the TMC-CRISPR test; the index test, the study used Seegene Allplex™ 2019-nCoV (Seegene) assay as the comparator reference gold standard throughout the study [9]. The study was conducted in a single blinded manner where the laboratory staff performing the index test and involved in results interpretation were blinded to the results of the comparator and routine RT-PCR tests. All the RT-PCR tests were conducted as per manufacturer's instructions, and the test cycle threshold (Ct) values generated by RT-PCR machine were reported. One or more targets with Ct values <40 were used to define a positive RT-PCR result. The comparator Seegene RT-PCR test results were interpreted based on the Ct values of N, E and RdRp genes.

## CRISPR assay

Reverse transcription PCR and amplification of cDNA were carried out to amplify S gene specific target sequences, in the RNA samples as per manufacturer's instruction [6]. Positive and negative controls were included in every batch. After completion of reverse transcription and cDNA amplification, tubes were transferred to post PCR area and TMC-CRISPR reaction was set up manually in dedicated biosafety cabinet (BSC). Master mix-S was prepared for required number of reactions by adding of dFNCas9 (1 μl/reaction), S gRNA (1 μl/reaction) and CRISPR buffer (3 μl/reaction). After amplification of cDNA, 5 μl of master mix -S was added to all PCR tubes and tubes were incubated at 37˚C for 10 minutes in a heat block. After 10 mins, tubes were removed from 37˚C; and 80 μl of pre-warmed LFS buffer was added into each tube. At the end one LFS card from the kit, was directly inserted into a strip of 8 tubes. Solution was allowed to migrate into the strip at room for not more than 2 mins and results were recorded manually as well as using Tata MD CHECK app- as per manufacturer's recommendations (**S1 Fig and S1 Table in S1 File**).

Nearly 91% of clinical specimens analysed in this study were freshly collected and processed on the same day without storage as recommended by the manufacturer. Prior to the start of the study, the estimated positivity of COVID-19 in Mumbai was 5%. However, during the course of the study, the positivity drastically dropped to 1% lowering the number of positive cases among fresh samples. Hence, frozen samples (n = 211; frozen for less than one month, constituting about 9% of all samples tested) with RT-PCR positive results were purposely selected for evaluation. This also helped in evaluating the performance of TMC-CRISPR test

on frozen samples, which is necessary in real life situations when the lab is unable to process all received samples on the same day. We did not gather sex/gender data of study participants under the assumption that this information does not affect study outcome. The diagnostic accuracy of the index test was evaluated based on sensitivity and specificity of the test as well as discordance analysis. Point estimates of sensitivity and specificity of the index test were reported in percentages along with 95% confidence intervals (CI) using Wilson's score method. Results were considered concordant if the results of the index test were similar to the comparator tests (positive or negative), and discordant if dissimilar. Discordance was assessed for index test results recorded through both manual and app-based techniques against the comparator RT-PCR test results. The statistical analysis was performed on all samples combined, including stratification by sample status (fresh vs frozen), presence or absence of COVID-19 symptoms in patients from whom the samples were collected (symptomatic vs asymptomatic) and Ct values. For analysis, the continuous Ct values for E, N and RdRp genes were categorised into $\leq 20$, 21–25, 26 to 30, 31 to 35 and $>35$. The median reaction turn-around time (TAT) was also compared for the index (both manual and app-based readings) and comparator tests. The TAT considered the RT-PCR process (conversion of RNA to cDNA, amplification of cDNA) and reporting of results. Data were managed and analysed using Microsoft Excel 360 and Stata 13SE (Stata Statistical Software: Release 13. College Station, TX: StataCorp LP. 2013) [10].

## Operational Feasibility Assessment (OFA)

The OFA followed a cross-sectional design that principally involved interviews with laboratory staff (N = 9) involved in sample processing using index test, results interpretation, data analysis and monitoring the project. This arm of the study was performed by an independent agency (NEERMAN) and involved initial and final assessments of the staff through two rounds of hybrid (telephonically and/or in person) interviews, one immediately after training when candidates performed one or two rounds of testing and the other one was taken at the end of the study after gaining more experience of testing. Written informed consent was received from each study participant prior to the interviews. Semi-structured questionnaire was used to collect data on infrastructure needs, operating conditions, HR requirements, training, user friendliness, quality control and other post-launch requirements of the index test. The OFA involved descriptive analyses of the data collected in the initial and final assessments with the laboratory staff.

## Results

To evaluate the performance of the index test, a total of 2,332 samples were analysed. The distribution of the clinical specimens based on sample type is provided in **Fig 1**. Of the total, 1,235, (53%) were throat swabs, 519 (22%) were nasopharyngeal swabs, 310 (13%) were nasal swabs, 167 (7%) throat plus nasopharyngeal swabs, 100 (4%) throat plus nasal swabs and one (0.04%) endotracheal aspirate.

The overall data consisted of fresh (N = 2,121, 91%) and frozen (N = 211, 9%) samples (**Table 1**). Total sample positivity by RT-PCR was 11.2%. Among fresh samples, 3.7% and among frozen samples 86.7% were positive by RT-PCR. In comparison, the total positivity of TMC-CRISPR test using manual reading method was six percent (2.6% among fresh and 41% among frozen samples). TMC-CRISPR app-based reading provided similar results. However, 31 (nearly 1.3%) samples were identified as inconclusive by app-based reading compared to only three (0.1%) samples being inconclusive by manual reading.

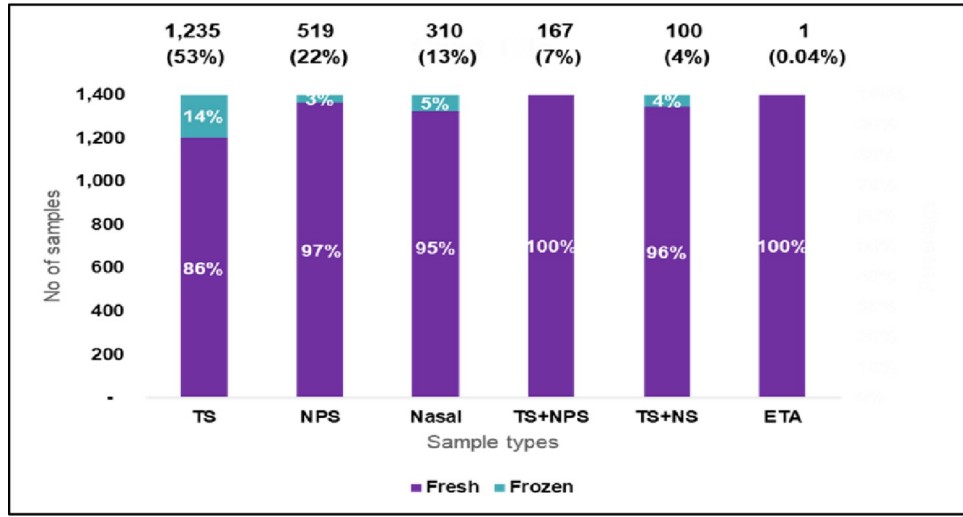

**Fig 1. Analysis of samples as per type of clinical specimen.** The distribution of the sample based on type of sample analysed is provided in Fig 1. Of the total, 1,235, (53%) were throat swabs, 519 (22%) were nasopharyngeal swabs, 310 (13%) were nasal swabs, 167 (7%) throat plus nasopharyngeal swabs, 100 (4%) throat plus nasal swabs and one (0.04%) endotracheal aspirate. Key- TS: Throat swab, NS: Nasal swab; NPS: Nasopharyngeal sample, ETA: Endo-tracheal aspirate.

Comparison of TMC-CRISPR test manually read results vs. RT-PCR results is presented in **Table 2**. Of the total, 140 (6%) were positive by TMC-CRISPR test and 261 (11%) were positive by RT-PCR. Overall sensitivity and specificity of the index test was estimated to be 44% and 99% respectively. Sensitivity was low in both fresh (41%) and frozen (45%) samples. Further, analysis of 95% CI (**Table 2**) showed overlapping CIs among fresh: 41.0 (30.6–52.2); and frozen: 45.4 (38.2–52.7) samples confirming no statistical difference in the sensitivity between these 2 types of specimens. Further, analysis of data using ICMR recommended Ct cut-off

**Table 1. Sample characteristics based on Seegene-Allplex™ 2019-nCoV RT-PCR test and TMC-CRISPR test results (manual & app-based reading), analysed at tertiary care centre—Mumbai, India, July–October, 2021.**

| | Total sample (n = 2,332) N (%) | Fresh samples (n = 2,121) N (%) | Frozen samples (n = 211) N (%) |
|---|---|---|---|
| **RT-PCR** | | | |
| Positive | 261 (11.2) | 78 (3.7) | 183 (86.7) |
| Negative | 2071 (88.8) | 2043 (96.3) | 28 (13.3) |
| **TMC-CRISPR (manual)** | | | |
| Positive | 140 (6.0) | 54 (2.6) | 86 (40.8) |
| Negative | 2189 (93.9) | 2064 (97.3) | 125 (59.2) |
| Inconclusive* | 3 (0.1) | 3 (0.1) | 0 (0) |
| **TMC-CRISPR (app-based)** | | | |
| Positive | 138 (5.9) | 52 (2.4) | 86 (40.8) |
| Negative | 2162 (92.7) | 2045 (96.4) | 117 (55.4) |
| Inconclusive* | 31 (1.3) | 23 (1.1) | 8 (3.8) |
| Invalid*** | 1 (0.1) | 1 (0.1) | 0 (0) |

*Presence of bands with very light intensities, difficult to interpret manually

**Band intensities were between 10–14.99

***There was no QR code present on the test strip rendering strip unreadable

**Table 2. Characteristics of the TMC-CRISPR test in comparison to Seegene-Allplex[TM] 2019-nCoV RT-PCR test based on samples tested at tertiary care centre—Mumbai, India, July–October, 2021.**

| Test diagnostic characteristic | All samples (N = 2,332) | Fresh samples (2,121) | Frozen samples (211) |
|---|---|---|---|
| Positive RT-PCR test results, N (%) | 261 (11.2) | 78 (3.7) | 183 (86.7) |
| Positive TMC-CRISPR (manual) test results, N (%) | 140 (6.0)[a] | 54 (2.6)[b] | 86 (40.7)[c] |
| Positive TMC-CRISPR (app based) test results, N (%) | 138 (5.9)[d] | 52 (2.5)[e] | 86 (40.8)[f] |
| Sensitivity of TMC-CRISPR (manual), % (95% CI) | 44.1 (38.1–50.1) | 41.0 (30.6–52.2) | 45.4 (38.2–52.7) |
| Specificity of TMC-CRISPR (manual), % (95% CI) | 98.8 (98.2–99.1) | 98.9 (98.4–99.2) | 89.3 (71.1–96.6) |
| PPV of TATA MD (manual), % (95% CI) | 82.1 (74.9–87.7) | 59.2(45.7–71.6) | 96.5 (89.6–98.9) |
| NPV of TATA MD (manual), % (95% CI) | 93.3 (92.2–94.3) | 97.8(97.0–98.3) | 20 (13.8–28.0) |
| Sensitivity of TMC-CRISPR (app-based), % (95% CI) | 44.4 (38.4–50.6) | 37.7 (27.5–49.0) | 47.4 (40.1–54.9) |
| Specificity of TMC-CRISPR (app-based), % (95% CI) | 98.7 (98.1–99.1) | 98.8 (98.2–99.2) | 89.3 (70.9–96.7) |
| PPV of TATA MD (app based), % (95% CI) | 81.2 (73.7–86.9) | 55.8 (42.0–68.7) | 96.5 (89.6–98.9) |
| NPV of TATA MD (app based), % (95% CI) | 93.5(92.4–94.4) | 97.6 (96.9–98.2) | 21.4 (14.8–29.8) |

[a]Distribution has 2,189 negative and 3 inconclusive results

[b]Distribution has 2,064 negative and 3 inconclusive results

[c]Distribution has 125 negative results

[d]Distribution has 2,162 negative, 31 inconclusive and 1 invalid results

[e]Distribution has 2,045 negative, 23 inconclusive, and 1 invalid results

[f]Distribution has 117 negative and 8 inconclusive results

PPV: Positive predictive value; NPV: Negative predictive value

value of ≤35 for RT-PCR positivity (**S2(a) and S2(b) Table in S2 File**) showed slightly increased sensitivity from 44% to 47%.

Detailed analysis of discordant results is shown in **Table 3**. Of the total positives, there were 146 (55.9%) samples positive by RT-PCR, negative by TMC-CRISPR manual and 140 (53.6%) positive by RT-PCR, negative by TMC-CRISPR app. Comparably, there were 25 samples (1.2%) negative on RT-PCR, positive on CRISPR manual, and 26 samples (1.3%) negative on RT-PCR positive on CRISPR app (**Table 3**). We also tried to analyse data based on presence or absence of clinical symptoms among COVID-19 suspects from whom specimens were collected for analysis (**Table 3**). Of the total, only 363 (16%) suspects were symptomatic and majority i.e., 1,970 (84%) suspects were asymptomatic. Overall, discordance between the comparator RT-PCR test and TMC-CRISPR test (manual) was found to be 56%. The discordance was 59% among fresh samples, 54% among frozen, 53% among asymptomatics, and 59% in individuals symptomatic for COVID-19 disease (**Table 3**). We observed similar results showing overall discordance of 54% using TataMD CHECK app.

When the reaction turn-around time (TAT) was compared, we found that the TMC-CRISPR reaction took more time than RT-PCR test, especially when app-based reading was involved (**Table 4**). The median time for completion of the RT-PCR process including conversion of RNA to cDNA, amplification of cDNA and obtaining results was 1.75 hours. The TMC-CRISPR test with manual reading took a median time of ~1.9 hrs whereas the TMC-CRISPR test with app-based reading took a median time of 3.1 hours.

Among RT-PCR positive patients, all 261 samples had a positive Ct value for both E (range:14 to 40) and N gene (13 to 40). Only 206/261 samples had an RdRp value reported. The RdRp target showed variations in the range of 0 to 43 (**S3 Table in S3 File**). Analysis of results based on RT-PCR Ct values showed a strong upward gradient in the discordance between RT-PCR and TMC-CRISPR test results (**Table 5**). Discordance between RT-PCR and TMC-CRISPR was less at low Ct value (high viral load). Discordance was found to be

**Table 3. Analysis of discordance between TMC-CRISPR test and RT-PCR test results[a], and between manual and app-based interpretation of TMC-CRISPR test results among sample and sub-samples tested.**

| | Overall | Sample status | | Symptom status[b] | |
|---|---|---|---|---|---|
| **RT-PCR vs TMC-CRISPR manual results** | Total sample (N = 2,332) | Fresh (N = 2,121) | Frozen (N = 211) | Symptomatic (N = 362) | Asymptomatic (N = 1,970) |
| RT-PCR positivity, N (%) | 261 (11.2) | 78 (3.7) | 183 (86.7) | 130 (35.9) | 131 (6.7) |
| TMC-CRISPR positivity (manual), N (%) | 140 (6.0) | 54 (2.6) | 86 (40.8) | 57 (15.7) | 83 (4.2) |
| TMC-CRISPR positivity (app-based), N (%) | 138 (5.9) | 52 (2.5) | 86 (40.8) | 56 (15.5) | 82 (4.2) |
| **RT-PCR vs TMC-CRISPR manual-based results** | | | | | |
| RT-PCR positive, TMC-CRISPR manual negative, n/d (%) | 146/261 (55.9) | 46/78 (59.0) | 100/183 (54.6) | 77/130 (59.2) | 69/131 (52.7%) |
| RT-PCR negative, TMC-CRISPR manual positive, n/d (%) | 25/2,071 (1.2) | 22/2,043 (1.1) | 3/28 (10.7) | 4/232 (1.7) | 21/1,839 (1.1%) |
| **RT-PCR vs TMC-CRISPR app-based results** | | | | | |
| RT-PCR positive, TMC-CRISPR app negative, n/d (%) | 140/261 (53.6) | 48/78 (61.5) | 92/183 (50.3) | 69/130 (53.1) | 71/131 (54.2) |
| RT-PCR negative, TMC-CRISPR app positive, n/d (%) | 26/2,071 (1.3) | 23/2,043 (1.1) | 3/28 (10.7) | 3/232 (1.3) | 23/1,839 (1.3) |
| **TMC-CRISPR Manual vs TMC-CRISPR app-based results** | | | | | |
| Manual positive and app negative, n/d (%) | 4/140 (2.8) | 4/54 (7.4) | 0/86 | 1/57 (1.8) | 3/83 (3.6) |
| manual negative and app positive, n/d (%) | 5/2,189 (0.2) | 5/2,064 (0.2) | 0/125 | 1/304 (0.3) | 4/1,885 (0.2) |

Abbreviations: RT-PCR = Reverse transcription–polymerase chain reaction; n = numerator; d = denominator; [a]False negative = RT-PCR positive and CRISPR negative; False positive = RT-PCR negative and CRISPR positive; [b] based on presence or absence of symptoms for COVID-19

increasing with increasing Ct values and decreasing viral load as described in Table 5. A similar trend was observed with N and RdRp gene targets.

## OFA study outcomes

Two rounds of interviews were completed for total nine study participants by the NEERMAN team. A summary of the OFA findings is described in **Table 6**. Of the nine participants (eight female and one male), five were lab technicians who conducted the test and the remaining 4 were study supervisors who were involved in overall work monitoring, result interpretation post analysis (both manual and app based), recording of reading/tests, and providing necessary support to the project as per requirement.

The average age of participants in the study was 30 years, with an age range of 23–37 years. At the time of both assessments, seven out of nine participants (78%) had more than twelve months of experience in handling RT-PCR for COVID-19, while the remaining two participants (22%) had an experience of less than 6 months.

As described in **Table 6**, in the final assessment, four out of nine participants stated a need for additional workspace (a separate post PCR area) and/or a dedicated bio safety cabinet to handle amplicons while setting up the TMC-CRISPR reaction. Furthermore, a light box (for

**Table 4. Comparison of Turn-around time between RT-PCR and TMC-CRISPR test.**

| Testing Type | Median Time taken |
|---|---|
| RT-PCR (post extraction, using RNA) | 105 minutes (1.75 hours) |
| TMC-CRISPR test (post extraction) with manual reading | 110 minutes (1.9 hours) |
| TMC-CRISPR test (post extraction) with app-based reading | 185 minutes (3.1hours) |

**Table 5. Analysis of discordance between TMC-CRISPR test and RT-PCR test results[a], across a range of Ct values among RT-PCR positive samples for target genes tested (N = 261).**

| RT PCR vs CRISPR results | Ct values: Target E gene (N = 261) | | | | |
|---|---|---|---|---|---|
| | Ct value <20 | Ct value 21 to 25 | Ct value 26 to 30 | Ct value 31 to 35 | CT value >35 |
| | (N = 32) | (N = 50) | (N = 77) | (N = 77) | (N = 25) |
| RT-PCR positive, TMC-CRISPR manual negative, N (%) | 5 (15.6%) | 16 (32.0%) | 40 (52.0%) | 63 (81.8%) | 22 (88.0%) |
| RT-PCR positive, TMC-CRISPR app negative, N (%) | 5 (15.6%) | 16 (32.0%) | 38 (49.3%) | 60 (77.9%) | 21 (84.0%) |
| **RT-PCR vs CRISPR results** | Ct values: Target N gene (N = 261) | | | | |
| | (N = 28) | (N = 49) | (N = 73) | (N = 85) | (N = 26) |
| RT-PCR positive, TMC-CRISPR manual negative, N (%) | 6 (21.4%) | 14 (28.6%) | 41 (562%) | 63 (74.1%) | 22 (84.6%) |
| RT-PCR positive, TMC-CRISPR app negative, N (%) | 5 (17.9%) | 15 (30.6%) | 38 (52.1%) | 61 (71.8%) | 21 (80.8%) |

Abbreviations: Ct = Cycle threshold. RT-PCR = Reverse transcription real time–polymerase chain reaction

[a]False negative = RT-PCR positive and CRISPER negative

optimal app-based result interpretation), specific models of smart phones and internet/wi-fi setup were essential for app-based data analysis.

In both assessments, eight out of nine participants confirmed that there was no need for additional training for specimen collection, RNA extraction and handling, and lab biosafety measures, beyond what was implemented in the COVID-19 RT-PCR lab. However, in the final assessment seven out of nine study staff cited requirement of at least one day training for setting up TMC-CRISPR test reaction, and interpretation of results. A change of opinion on the complexity / ease of conducting the TMC-CRISPR test was observed between initial and final assessments. In the initial assessment (Table 6), eight out of nine participants believed the difficulty levels of TMC-CRISPR test and RT-PCR were similar, whereas in the final assessment after gaining experience and testing more batches of samples, four out of nine participants mentioned that compared to RT-PCR testing, the TMC-CRISPR test was more challenging to undertake. Similarly, in the initial assessment, four out of nine participants found the difficulty level for result interpretation of TMC-CRISPR test and RT-PCR to be the same. Three participants stated that the result interpretation for TMC-CRISPR test was less difficult compared to RT- PCR while two out of nine participants mentioned result interpretation was more difficult for the TMC-CRISPR test compared to the RT-PCR test. However, there was no change in their opinion at the end of final assessment. Further, the participants were inquired about the accuracy of app-based interpretation in comparison to manual interpretation of the TMC-CRISPR test. In the initial assessment, four expressed a favourable opinion, stating that the app's AI interpretation proved to be more accurate and particularly beneficial in determining borderline results. Among the remaining five, participants, two there is no difference between app based and manual results, and remaining two shared their view as that the app's AI interpretation was less accurate compared to manual interpretation. Meanwhile, only one participant stated that they were not familiar with the AI-based interpretation. Only, one/nine participants changed their responses from initial to final assessment.

As just four of the study supervisors were involved in final results interpretation, based on their responses during the final assessment, we conclude that manual and app-based result interpretation of the index test, are both more challenging than RT-PCR testing. Two of the supervisors, involved in the post analytical processes, mentioned certain drawbacks of using the Tata MD CHECK app that include erroneous results, internet issues, challenges in updating app regularly, a shadow on the results band or faint bands that cause interpretation errors. In the final assessment, all participants mentioned that the current version of the CRISPR based Tata MD CHECK may not be considered a suitable alternative to RT-PCR.

**Table 6. Findings of CRISPR based TMC-CRISPR test for COVID-19 compared to RT-PCR test, initial vs final OFA.**

| Variables | Initial assessment (n = 9) | Final assessment (n = 9) |
|---|---|---|
| | N (%) | N (%) |
| **Age,Mean, (SD)** | 30 (4.3) | 30 (4.3) |
| **Sex** | | |
| Females (%) | 8 (89%) | 8 (89%) |
| Males (%) | 1 (11%) | 1 (11%) |
| **Designation** | | |
| Lab technician (%) | 5 (55%) | 5 (55%) |
| Supervisor (%) | 4 (45%) | 4 (45%) |
| **Experience with RT-PCR testing for COVID-19** | | |
| < 3 months | 1 (11%) | 0 |
| 3–6 months | 1 (11%) | 1 (11%) |
| 7–12 months | 0 | 1 (11%) |
| >12 months | 7 (78%) | 7 (78%) |
| **Experience with RNA extraction for COVID-19** | | |
| < 3 months | 2 (22%) | 0 |
| 3–12 months | 0 | 2 (22%) |
| >12 months | 7 (78%) | 7 (78%) |
| **Requirement for additional workspace and equipment for TMC-CRISPR** | | |
| Yes | 1 (11%) | 4 (44%) |
| No | 8 (89%) | 5 (55%) |
| **Additional biosafety requirements to conduct TMC-CRISPR test** | | |
| Yes | 1 (11%) | 1 (11%) |
| No | 8 (89%) | 8 (89%) |
| **Additional precautions/training requirement for handling extracted RNA for conducting TMC-CRISPR test compared to RT—PCR** | | |
| Yes | 1 (11%) | 1 (11%) |
| No | 8 (89%) | 8 (89%) |
| **Responses on first interaction with TMC-CRISPR test** | | |
| Easy, only a user manual is required | 9 (100%) | 8 (89%) |
| Difficult, need onsite training | 0 | 1 (11%) |
| **Difficulties encountered with TMC-CRISPR test** | | |
| No difficulties faced | 5 (4L, 1S) | 6 (4L, 1S) |
| Difficulties faced in test procedures | 4 (3S) | 3 (1S) |
| **Training requirement (in days)** | | |
| One Day | 4 (45%) | 7 (78%) |
| Two Days | 3 (33%) | 1 (11%) |
| > 2 Days | 2 (22%) | 1 (11%) |
| **Was the average turnaround time of TMC-CRISPR test similar to RT–PCR** | | |
| Yes | 3 (33%) | 0 |
| No | 6 (67%) | 9 (100%) |
| **Ease of conducting TMC-CRISPR test compared to RT—PCR** | | |
| Less difficult | 1 (11%) | 0 |
| Same | 8 (89%) | 5 (55%) |
| More difficult | 0 | 4 (45%) |
| **Ease of result interpretation of TMC-CRISPR compared to RT—PCR** | | |

(*Continued*)

**Table 6.** (Continued)

| Variables | Initial assessment (n = 9) | Final assessment (n = 9) |
|---|---|---|
| Same as RT-PCR | 4 (45%) | 4 (45%) |
| Less difficult than RT-PCR | 3 (33%) | 3 (33%) |
| More difficult than RT-PCR | 2 (22%) | 2 (22%) |
| **TMC-CRISPR result Interpretation accuracy (App vs Manual reading)** | | |
| Same | 2 (22%) | 2 (22%) |
| App more accurate | 4 (45%) | 3 (33%) |
| App less accurate | 2 (22%) | 3 (33%) |
| DO not know/Can't say | 1 (11%) | 1 (11%) |
| **Can TMC-CRISPR test be used as an alternative to RT-PCR for COVID-19** | | |
| Yes | 3 (33%) | 0 |
| No | 5 (55%) | 9 (100%) |
| Can't say | 1 (11%) | 0 |

## Discussion

In the COVID-19 pandemic, CRISPR based tests have received substantial attention for nucleic acid detection due to their simplicity, speed, manufacturer-stated good sensitivity, and specificity [11, 12]. We examined performance characteristics and OFA of TMC-CRISPR against ICMR approved RT-PCR assay at VRDL laboratory of KEM hospital with few limitations. The current study was conducted only in one lab with a small number of staff (n = 9) interviewed in the OFA. Nearly 84% patients included in this study were asymptomatic and during the period of the study, the positivity rate varied widely (positivity decreased from 17% to 1%) due to a "trough" in the C COVID-19 pandemic in the city of Mumbai.

The low TMC-CRISPR assay sensitivity (under 50%) could be attributed to multiple reasons. The TMC-CRISPR assay aims to detect only one SARS-CoV-2 gene—"S" gene that is prone to more frequency of mutations and, it uses strip-based detection method that may be less sensitive compared to fluorescence-based approaches used in RT-PCR. There are limited reports where lateral flow paper strip method was utilised for detection of SARS-CoV-2 after amplification of cDNA. Agarwal et. al. utilized loop-mediated isothermal amplification (LAMP) followed by lateral flow–based nucleic acid detection of SARS-CoV-2 and reported sensitivity of 77% and LOD of $3.9 \times 10^3$ that is much lower that RT-PCR sensitivity (50–100 copies/mL) [13].

As per the manufacturer's instruction manual [6] the limit of detection (LOD) of the TMC-CRISPR test, is 10–100 copies of the target genetic sequence when the assay is performed under controlled laboratory conditions by well-trained personnel and under real life diagnostic conditions it can go up to 500–1000 copies of the viral genome. In contrast, the RT-PCR assays used in this study have more than one gene targets with much lower limit of detection i.e., ≤100 copies of viral genome per reaction [14] (S4 Table in S4 File). Further, all the RT-PCR assays utilised in this study have in-build internal controls to provide assurance that clinical specimens are successfully amplified and detected. In case of inhibition, internal control helps suggesting repat sample for testing. In contrast, the current version of TMC-CRISPR assay does not have in-built internal controls to verify the reaction success of each sample. Therefore, in case of inhibition of reverse transcription PCR, CRISPR reaction will also not take place, affecting the final test outcome as false negative results.

Significant discrepancies were noted in the TMC-CRISPR results (both manual and app-based) compared to RT-PCR, regardless of the sample type or the presence of clinical

symptoms. A detailed examination of the discordant results, considering the Ct values for E, N, and RdRp gene targets obtained from RT-PCR, revealed a discernible pattern. As the Ct values increased, indicating lower viral loads, the discordance between TMC-CRISPR and RT-PCR results demonstrated an evident upward trend (**Table 5**). This crucial finding emphasizes the importance of conducting extensive testing on a larger number of clinical specimens containing low viral loads, particularly those near or below LOD of newer diagnostic tests, during the validation and clinical evaluation phases. A thorough assessment of such samples is essential before widespread implementation of these diagnostic tests in real-world scenarios.

OFA outcomes highlighted a need of proper unidirectional laboratory set up to perform TMC-CRISPR test and equipment like refrigerated centrifuge for RNA extraction. Further, TMC-CRISPR test has more complex manual steps after amplification of cDNA and necessitates training for careful handling of amplified products in separate area to avoid amplicon contamination and false positive results. Opening of each tube to add the CRISPR reagent, insert the paper strips and reading of the strips manually require additional time and attention. To use TMC-app, band images and data are to be uploaded onto the Tata MD CHECK website for result interpretation, a step that can be more time consuming (than manual reading) in case of poor internet connectivity. Issues such as erroneous results, difficulty in interpretation of faint bands, data safety and security were highlighted as major challenges and concerns with the app-based readings. Moreover, app-based result interpretation of TMC-CRISPR test requires a light box for band visualisation, a smart phone with certain specifications and an active internet connection at an additional cost, further limiting the use of this technique.

In summary, the TMC-CRISPR assay was initially designed and promoted as a less complex and more field-friendly alternative to RT-PCR for COVID-19 testing. It offered the advantage of not requiring an RT-PCR machine and a unidirectional work setup, potentially making it a cost-effective option. However, the results of this study suggest that the current version of the TMC-CRISPR test cannot replace RT-PCR due to its poor sensitivity. This limitation hinders its reliability for clinical decision-making. Nonetheless, there is potential for improvement, as the test can be further refined to reduce manual steps, enhance ease of use, and decrease TAT. Future work may focus on assay redesigning and development, incorporating improved targets and a more sensitive detection methods for increasing the sensitivity and accessibility for point of care use. Further, to be utilised as a diagnostic test in populations like asymptomatic patients with low viral loads, extensive validation of claimed LOD under real field conditions is recommended. It is only post such improvements, future version of more rapid, automated and sample-to-result CRISPR-based diagnostic test might prove impactful at peripheral primary healthcare settings.

## Supporting information

**S1 File. This is supporting file for S1 Fig and S1 Table.**
(PDF)

**S2 File. This is supporting file for S2(a) and S2(b) Tables.**
(PDF)

**S3 File. This is supporting file for S3 Table and supporting Ct value information.**
(PDF)

**S4 File. This is supporting file for S4 Table.**
(PDF)

**S5 File. Other data files (Other data file 1 and 2).**
(PDF)

## Acknowledgments

We thank NEERMAN team for their help and support in conducting feasibility assessment.

## Author Contributions

**Conceptualization:** Shubhada Shenai, Gita Nataraj, Sanjay Sarin, Sarabjit S. Chadha.

**Data curation:** Gita Nataraj, Akhil S. Thekke Purakkal.

**Formal analysis:** Shubhada Shenai, Gita Nataraj, Minal Jinwal, Akhil S. Thekke Purakkal, Rajashree Sen, Trupti Mathure, Sarabjit S. Chadha.

**Funding acquisition:** Sanjay Sarin, Sarabjit S. Chadha.

**Investigation:** Gita Nataraj, Minal Jinwal.

**Methodology:** Shubhada Shenai, Akhil S. Thekke Purakkal, Nayana Ingole, Sanjay Sarin.

**Project administration:** Shubhada Shenai, Gita Nataraj, Rajashree Sen, Sanjay Sarin, Sarabjit S. Chadha.

**Resources:** Rajashree Sen.

**Supervision:** Shubhada Shenai, Gita Nataraj, Minal Jinwal, Nayana Ingole, Trupti Mathure, Sarabjit S. Chadha.

**Visualization:** Shubhada Shenai, Gita Nataraj, Minal Jinwal, Akhil S. Thekke Purakkal, Nayana Ingole, Trupti Mathure.

**Writing – original draft:** Shubhada Shenai.

**Writing – review & editing:** Shubhada Shenai, Gita Nataraj, Minal Jinwal, Akhil S. Thekke Purakkal, Rajashree Sen, Nayana Ingole, Sanjay Sarin, Sarabjit S. Chadha.

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
