## [Decision Letter · Decision Letter 0]

6 Jul 2023

PONE-D-23-14455Performance characteristics and operational feasibility assessment of a CRISPR based Tata MD CHECK diagnostic test for SARS-CoV-2 (COVID-19)PLOS ONE

Dear Dr. Shenai,

Thank you for submitting your manuscript to PLOS ONE. After careful consideration, we feel that it has merit but does not fully meet PLOS ONE’s publication criteria as it currently stands. Therefore, we invite you to submit a revised version of the manuscript that addresses the points raised during the review process.

We look forward to receiving your revised manuscript.

Kind regards,

Ruslan Kalendar

Academic Editor

PLOS ONE

Journal Requirements:

"This work was supported, in whole or in part, by the Bill & Melinda Gates Foundation [INV-025422]. Under the grant conditions of the Foundation, a Creative Commons Attribution 4.0 Generic License has already been assigned to the Author Accepted Manuscript version that might arise from this submission. Our sincere thanks to NIRMAN team for all their help and support in conducting feasibility assessment. "

"The authors received no specific funding for this work"

6. Please include a copy of Table 7 which you refer to in your text on page 12.

Reviewers' comments:

Reviewer's Responses to Questions

**Comments to the Author**

1. Is the manuscript technically sound, and do the data support the conclusions?

Reviewer #1: Yes

Reviewer #2: Yes

2. Has the statistical analysis been performed appropriately and rigorously? 

Reviewer #1: N/A

Reviewer #2: Yes

3. Have the authors made all data underlying the findings in their manuscript fully available?

Reviewer #1: Yes

Reviewer #2: No

4. Is the manuscript presented in an intelligible fashion and written in standard English?

Reviewer #1: Yes

Reviewer #2: Yes

5. Review Comments to the Author

Reviewer #1: 

In this work, the authors perform a validation of the sensitivity and specificity of a CRISPR-based test vs the gold-standard PCR. This report shows quite convincingly that the named test does not work, despite having been approved for Emergency Use. As such, I think it is important to publish these data. Additionally, the authors provide a small survey about the ease of use on the test, which is quite original. In general, the table of sensitivity, specificity, positive predictive value and negative predictive value of their test should be in the main file, rather than the Supplemental. It is more informative that % samples and quite important when evaluating the given performance of a test. Can the authors provide the thresholds under which different tests are approved according to their healthcare system? They may change depending on their use (e.g. clinical diagnostics vs population screening; PCR vs lateral flow). More detailed comments are below:

Line 51: the description of TMC-CRISPR includes ‘reverse transcription PCR’ – I would highlight that the main difference with standard real time PCR is how the detection of the amplified product is done. At the beginning of the paper, the main premise of using a lateral flow type test is that RT-PCR is complex and expensive to perform.

Ct values are difficult to interpret since thresholds vary from lab to lab – do the authors have an idea of how many copies per mL do Ct = 20 correspond to?

Lines 260-261 and 271 appear to contradict each other. Did the participants change or not change their opinion? Does line 271 relate to interpretation of the data? Please clearly state so, there were shifts in opinions as explained by the authors, which are important findings.

In Discussion (check typo, Discussions), I am missing SHERLOCK as a pioneer system to detect SARS-CoV-2 using CRISPR.

Please be consistent about the annotation ‘COVID-19’ and not ‘Covid-19’.

N=9 is not that a small number for a laboratory – for a survey it may be, so I would put that as a limitation although the data is quite clear.

Does the TMC-CRISPR assay include a control line? Including a picture of the assay would be quite helpful if possible.

The authors mention sensitivity of copies- is that per mL sample? Lateral flow assays have lower thresholds to be considered under UK regulations. Is that the case? In any case, a Ct of 20 should be in the range of millions of copies per mL.

Please refer to your figures and tables when you discuss them.

I am also missing some more literature on the general performance of lateral flow assays.

Line 316: what does ‘gradient’ mean? Do the authors mean that with higher Ct values per RT-qPCR they observed more false negatives in the TMC-CRISPR? If so, please clearly say so.

Line 323: I do not understand the argument about a refrigerated centrifuge for RNA extraction, if the gold-standard RT-qPCR worked with the RNA extracted without such equipment. Do the authors imply that there is RNA degradation in their samples – or that it is so extensive that the test does not work? There is an RT-PCR step also performed, it is the detection of the amplicons that changes.

Line 339: what is ‘TAT’

I am not sure how ‘extensive validation’ will help in this case. If the test is not sensitive enough, it requires re-designing. As it stands, the authors have shown quite convincingly that it does not work.

Is there a possibility that the test was designed and approved for the original variant and it does not work with the tested variant?

Reviewer #2: 

Shenai S et al assessed and described the performance of a CRISPR based Tata MD CHECK diagnostic test for SARS-CoV-2( COVID -19) and feasibility of the test’s operations in KEM hospital Mumbai. Based on their findings they did not recommend the use of the test.

Comments

1.Table 2 . Although the negative results and inconclusive are mentioned in the legends. I propose making a simpler table of TMC-CRISPR vs Seegane RT-PCR showing the positives and negatives in the table.

2.Study limitation.

Consider including the use of different sample types ie nasopharyngeal, throat swab and nasal swab and endotracheal aspirates in the analysis as a potential source of bias since the different sample types may have different performances .

3. Line 322. The authors mention OFA highlights need for unidirectional laboratory setup to perform TMC-CRISPR test. Since this is also required in the RT-PCR I think this point should be excluded from the manuscript.

6. PLOS authors have the option to publish the peer review history of their article (what does this mean?). If published, this will include your full peer review and any attached files.

Reviewer #1: No

Reviewer #2: No

While revising your submission, please upload your figure files to the Preflight Analysis and Conversion Engine (PACE) digital diagnostic tool, https://pacev2.apexcovantage.com/. PACE helps ensure that figures meet PLOS requirements. To use PACE, you must first register as a user. Registration is free. Then, login and navigate to the UPLOAD tab, where you will find detailed instructions on how to use the tool. If you encounter any issues or have any questions when using PACE, please email PLOS at figures@plos.org. Please note that Supporting Information files do not need this step.<quillbot-extension-portal></quillbot-extension-portal>

---

## [Author Response · Author response to Decision Letter 0]

24 Aug 2023

Response to comments received from Reviewers: -

• Reviewer # 1

# Comment # 1

a. In general, the table of sensitivity, specificity, positive predictive value and negative predictive value of their test should be in the main file, rather than the Supplemental. It is more informative than % samples and quite important when evaluating the given performance of a test. 

b. Can the authors provide the thresholds under which different tests are approved according to their healthcare system? They may change depending on their use (e.g. clinical diagnostics vs population screening; PCR vs lateral flow).

Response:

a. Thank you for the feedback. We have revised the table 2 in the main file which presents estimates and 95% CIs for sensitivity, specificity, positive predictive value and negative predictive value for TATA MD (manual) and TATA MD (app based) readings. 

b. Validation of SARS-COV2 diagnostic commodities: ICMR acceptance Criteria is described below 

RT-PCR Kit Sensitivity: 95% and above

 Specificity: 99% and above 

RNA Extraction Kit At least 95% concordance among positive 

 At least 90% concordance among negative samples 

 > 95 % samples showing amplification in internal control 

VTM 100% concordance among spiked samples 

 100% samples showing amplification in internal control 

Antibody Rapid Kit Sensitivity: 90% and above 

 Specificity: 99% and above 

ELISA / CLIA Kit IgM: Sensitivity - 90% and above 

 Specificity- 99% and above 

 IgG: Sensitivity- 90% and above 

 Specificity- 95% and above 

Notes: 

1. The cut-offs have been decided after deliberation in ICMR Expert Group with the Drug Controller General of India (DCGI). 

2. In case of TMC-CRISPR, manufacturer’s claimed sensitivity of 96.1% and specificity of 98.6% on 149 extracted RNA samples (77 positive, 72 negative) at a reputed 3rd party molecular biology lab and research institute (Reference 6 main manuscript). 

# Comment # 2

Line 51: the description of TMC-CRISPR includes ‘reverse transcription PCR’ – I would highlight that the main difference with standard real time PCR is how the detection of the amplified product is done. At the beginning of the paper, the main premise of using a lateral flow type test is that RT-PCR is complex and expensive to perform.

Response

We appreciate the reviewer for bringing up this important point. The TMC-CRISPR assay was initially designed and promoted as a potential replacement for RT-PCR, with the advantage of being less complex and more suitable for field use especially in remote locations in the country. The key features highlighted were the ability to work without the need for an RT-PCR machine and a unidirectional work setup, making it a potentially cost-effective and portable option for COVID-19 testing. However, upon close examination of the assay's principles and work process, it became apparent that further evaluation of its feasibility in a real-world RT-PCR laboratory setup was necessary. The decision to conduct the feasibility evaluation in an RT-PCR laboratory setting was driven by the need to thoroughly assess the assay's performance and reliability, particularly in a controlled and well-established diagnostic environment. 

# Comment # 3

Ct values are difficult to interpret since thresholds vary from lab to lab – do the authors have an idea of how many copies per mL do Ct = 20 correspond to?

Response:

We agree with the reviewer’s comment that there is always slight variation in Ct values as thresholds vary from lab to lab. Further, to determine the number of copies per mL corresponding to a Ct value of 20, we need a standard curve or a known reference sample with a known concentration of the target nucleic acid. We have not done standard curve analysis as that was not the aim for this study. Kit validation was carried out using known positive and negative samples. However, looking at the limit of detection of Seegene RT -PCR, and considering 3-4 Ct difference = 1 log difference, Ct = 20 might be ~ ≥106-7 copies (10,00,000 -1,00,00,000 copies) per reaction. 

# Comment # 4

Lines 260-261 and 271 appear to contradict each other. Did the participants change or not change their opinion? Does line 271 relate to interpretation of the data? Please clearly state so, there were shifts in opinions as explained by the authors, which are important findings.

Response

As outlined in table 6, the interpretation of TMC-CRISPR test results compared to RT-PCR results was discussed in lines 260-261. Notably, all participants maintained their initial opinions throughout the final assessment, indicating that there were no changes in the outcomes between the initial and final assessments.

Additionally, line 271 pertains to the interpretation of data obtained from the app-based system versus the manual method. Subsequent revisions were made to the manuscript based on the findings from this analysis presented in table 6. 

# Comment # 5

In Discussion (check typo, Discussions), I am missing SHERLOCK as a pioneer system to detect SARS-CoV-2 using CRISPR.

Response: 

Thank you for pointing out this typo error. We have addressed the suggested corrections and made the necessary changes.

# Comment # 6

Please be consistent about the annotation ‘COVID-19’ and not ‘Covid-19’.

Response: 

Thank you for bringing this error to our attention. We have carefully reviewed the manuscript and made the necessary corrections to ensure consistency throughout.

# Comment # 7

N=9 is not that a small number for a laboratory – for a survey it may be, so I would put that as a limitation although the data is quite clear.

Response

Agree. We highlighted this drawback at the beginning of discussion section.

# Comment # 8

Does the TMC-CRISPR assay include a control line? Including a picture of the assay would be quite helpful if possible.

Response 

Yes, the TMC-CRISPR assay incorporates a control line to ensure proper functionality. To provide clarity on the results interpretation of this assay, we have included a standard picture and table, offering a comprehensive understanding of the outcomes in supporting documents. 

# Comment # 9

The authors mention sensitivity of copies- is that per mL sample? Lateral flow assays have lower thresholds to be considered under UK regulations. Is that the case? In any case, a Ct of 20 should be in the range of millions of copies per mL. Please refer to your figures and tables when you discuss them.

Response

Please note that the limit of detection (LOD) of each assay is mentioned as copies per reaction. We agree with reviewer’s comment that lateral flow assays have lower thresholds and that could be one of the reasons of higher limit of detection and lower sensitivity of the TMC CRISPR assay.

Necessary changes were made referring figures and tables in the text as suggested.

# Comment # 10

I am also missing some more literature on the general performance of lateral flow assays.

Response

The detection of amplified PCR products using the lateral flow paper strip method is generally considered as a rapid, cost effective and convenient approach to visualize the presence of a specific target sequence in the PCR reaction in low-income settings. The sensitivity of such a paper strip-based lateral flow assays for PCR product detection can vary depending on various factors, including the target analyte, the design of the assay, and the detection method employed etc. There are limited reports where lateral flow paper strip method was utilised for detection of SARS-CoV-2 after amplification of cDNA. Agarwal et. al. utilized loop-mediated isothermal amplification (LAMP) followed by lateral flow–based nucleic acid detection of SARS‑CoV‑2 and reported sensitivity of 77% and LOD of 3.9 x 103 that is much lower that RT-PCR sensitivity (50-100 copies/mL). Relevant reference has been added in the main manuscript. 

# Comment # 11

Line 316: what does ‘gradient’ mean? Do the authors mean that with higher Ct values per RT-qPCR they observed more false negatives in the TMC-CRISPR? If so, please clearly say so.

Response

Gradient meaning a slope (incline or decline trend). Yes. We did observe more false negatives (descending trend) in TMC-CRISPR results with increasing RT-PCR Ct values. 

# Comment # 12

Line 323: I do not understand the argument about a refrigerated centrifuge for RNA extraction, if the gold standard RTqPCR worked with the RNA extracted without such equipment. Do the authors imply that there is RNA degradation in their samples – or that it is so extensive that the test does not work? There is an RT-PCR step also performed, it is the detection of the amplicons that changes.

Response:

In the discussion, it is essential to highlight that RNA extraction from clinical specimens typically necessitates the use of a refrigerated centrifuge. The manufacturer claimed that the test could be utilized in remote locations as it does not require RT-PCR machine; however, it is crucial to consider that such areas might lack access to a refrigerated centrifuge for RNA extraction. Subsequently, performing the test in the field and remote locations might not be feasible, posing a significant challenge to its practical implementation in these settings. 

# Comment # 13

Line 339: what is ‘TAT’

Response: 

"TAT" is an abbreviation for "turnaround time." Please refer to line 215 for a detailed explanation where the full form and its corresponding abbreviation are provided.

# Comment # 14

I am not sure how ‘extensive validation’ will help in this case. If the test is not sensitive enough, it requires re-designing. As it stands, the authors have shown quite convincingly that it does not work. Is there a possibility that the test was designed and approved for the original variant, and it does not work with the tested variant?

Response:

We are sincerely grateful to the reviewer for bringing up this significant aspect. In the case of TMC-CRISPR assay, the manufacturers must undertake a diligent process of assay redesign and development, incorporating improved targets and a more sensitive detection method than LAS. However, the pivotal step following the development phase is to conduct an extensive validation of the assay. This involves testing a substantial number of samples with lower viral loads, approaching the limit of detection (LOD) of the assay. Such thorough validation is indispensable to ensure the assay's effectiveness and suitability for use in real-world field conditions, which can vary considerably. 

Reviewer # 2

# Comment # 1

Table 2. Although the negative results and inconclusive are mentioned in the legends. I propose making a simpler table of TMC-CRISPR vs Seegene RT-PCR showing the positives and negatives in the table.

Response: 

Thank you for the comment. Details of TMC-CRISPR vs Seegene RT-PCR showing positives, negatives, inconclusive and invalid have already been provided in table 1. We hope this table meets the suggested distribution. 

# Comment # 2

Study limitation. Consider including the use of different sample types ie nasopharyngeal, throat swab and nasal swab and endotracheal aspirates in the analysis as a potential source of bias since the different sample types may have different performances.

Response: 

This study encompasses a comprehensive range of sample types, including throat swabs, nasal swabs, nasopharyngeal swabs, and a combination of throat + Nasal swabs and throat + nasopharyngeal swabs, etc. as depicted in figure 1. Our analysis delved deeply into the outcomes based on these distinct sample types and we observed consistent specificity across all types. However, it is noteworthy that sensitivity exhibited some variability. Further, it’s important to mention that the sample size for the combined throat + nasal swabs subset was relatively smaller, rendering the interpretation of a sensitivity score of 100 impractical (not interpretable). As a result, we have omitted this particular aspect from our discussion for the sake of clarity and accuracy.

# Comment # 3

 Line 322. The authors mention OFA highlights need for unidirectional laboratory setup to perform TMC-CRISPR test. Since this is also required in the RT-PCR I think this point should be excluded from the manuscript.

Response: 

The response is similar to the one highlighted in response to comment 2 from reviewer 1. We appreciate the reviewer for bringing up this important point. The TMC-CRISPR assay was initially designed and promoted as a potential replacement for RT-PCR, with the advantage of being less complex and more suitable for field use especially in the remote locations in the country. The key features highlighted by the manufacturer were the ability to work without the need for an RT-PCR machine and a unidirectional RT-PCR work setup, making it a potentially cost-effective and portable option for COVID-19 testing. However, upon close examination of the assay's principles and work process, it became apparent that further evaluation of its feasibility in a real-world RT-PCR laboratory setup was necessary. Hence, we think this point is important to discuss in the discussion section.

Please note that there is a correction in Authors name A.S. Thekkepurallar should be written as Akhil S. ThekkePurakkal. Furthermore, it should be noted that authors M. Jinwal and Akhil S. ThekkePurakkal have made an equivalent contribution to this manuscript, thereby sharing equal third authorship. It is important to highlight that this acknowledgment does not alter the established sequence of author listing.

Once again, thank you for your efforts in helping us to improve this manuscript, and we are grateful for your continued support.

Note: Detailed response letter answering all the queries raised by academic editor and both reviewers is attached.

---

## [Editor Report · Decision Letter 1]

25 Aug 2023

Performance characteristics and operational feasibility assessment of a CRISPR based Tata MD CHECK diagnostic test for SARS-CoV-2 (COVID-19)

PONE-D-23-14455R1

Dear Dr. Shenai,

We’re pleased to inform you that your manuscript has been judged scientifically suitable for publication and will be formally accepted for publication once it meets all outstanding technical requirements.

Kind regards,

Ruslan Kalendar

Academic Editor

PLOS ONE

---

## [Editor Report · Acceptance letter]

7 Sep 2023

PONE-D-23-14455R1 

Performance characteristics and operational feasibility assessment of a CRISPR based Tata MD CHECK diagnostic test for SARS-CoV-2 (COVID-19) 

Dear Dr. Shenai:

I'm pleased to inform you that your manuscript has been deemed suitable for publication in PLOS ONE. Congratulations! Your manuscript is now with our production department. 

Kind regards, 

on behalf of

Professor Ruslan Kalendar 

Academic Editor

PLOS ONE